# Reading the Score of the Air—Change in Airborne Microbial Load in Contrast to Particulate Matter during Music Making

**DOI:** 10.3390/ijerph19169939

**Published:** 2022-08-12

**Authors:** Birte Knobling, Gefion Franke, Lisa Beike, Timo Dickhuth, Johannes K. Knobloch

**Affiliations:** Institute for Medical Microbiology, Virology and Hygiene, University Medical Center Hamburg-Eppendorf, 20251 Hamburg, Germany

**Keywords:** indoor air quality, pharyngeal flora, orchestra, particulate matter, displacement ventilation

## Abstract

The potential impact of music-making on air quality around musicians was inferred at the outset of the SARS-CoV-2 pandemic from measurements on individual musical instruments and from theoretical considerations. However, it is unclear to what extent playing together in an orchestra under optimal ventilation conditions really increases infection risks for individual musicians. In this study, changes in indoor air quality were assessed by measuring common parameters, i.e., temperature, relative humidity, and carbon dioxide, along with particle counting and determining the presence of airborne pharyngeal bacteria under different seating arrangements. The study was conducted in cooperation with a professional orchestra on a stage ventilated by high volume displacement ventilation. Even with a full line-up, the particle load was only slightly influenced by the presence of the musicians on stage. At the same time, however, a clear increase in pharyngeal flora could be measured in front of individual instrument groups, but independent of seat spacing. Simultaneous measurement of various air parameters and, above all, the determination of relevant indicator bacteria in the air, enables site-specific risk assessment and safe music-making even during a pandemic.

## 1. Introduction

Early in the COVID-19 pandemic, a number of transmission events could be attributed to music making, particularly singing together in a choir [1,2]. Subsequently, in addition to studies of aerosol release from singing [3], the amount of aerosol and droplet release from wind instruments [4] and their potential to travel greater distances was investigated. Furthermore, the ability of the new SARS-CoV-2 virus to persist for some time in aerosol particles and on inanimate surfaces as an infectious agent [5] has been demonstrated.

These findings influenced health care authorities’ containment measures of the SARS-CoV-2 pandemic and led to considerable restrictions in the cultural scene, especially at the beginning of the pandemic. Orchestral musicians in particular were almost completely restricted in their professional practice for a long time by the establishment of rules regarding upper limits on the number of people in the room and minimum distances between musicians on stage [6].

Many different studies have focused on individual aspects of music making, especially with wind instruments, in order to assess possible transmission risks. In some cases, slightly different results were obtained for individual instrument groups and instruments. By comparing the release of particles during breathing and speaking with the release during music making [4,7], risk profiles for different instruments were suggested. However, playing particularly loud notes with a brassy sound was not associated with a particular increased release of particles [8].

The distribution of particles indoors is influenced, among others, by the size of the room, the number of people present, the release of particles as part of their activity, as well as the type and extent of ventilation [9,10]. There are only a few studies investigating the dispersion of particles in concert halls. Recently, the risk of transmission of pathogens in the audience of a concert hall with displacement ventilation system was evaluated by measuring the aerosol and CO_2_ concentration in the surrounding area of a mannequin dummy [11]. However, in the context of the performance of orchestral music, the risk for the musicians on a stage is of particular interest. Previous data primarily represent measurements with single instruments or modeling of the dispersion of potentially infectious droplets and aerosols [12].

The aim of the present study was to further determine the potential transmission risk for the entire orchestra on stage. Therefore, the influence of seating distances and the number of musicians involved on indoor air quality was investigated during repeated performances of the same 20-min piece of music by a professional classical orchestra on a stage ventilated with a displacement ventilation system. CO_2_ content, air humidity, temperature, and particulate matter were measured as well as the detection and microbiological differentiation of airborne indicator bacteria at various positions within the orchestra.

## 2. Materials and Methods

The experiments were performed contiguously in one day in cooperation with a professional orchestra on a stage in northern Germany. Participation was voluntary for the orchestra members, provided that they tested negative for SARS-CoV-2 by polymerase chain reaction (PCR) from throat swabs on the day before.

The ventilation technology of the stage is based on the principle of displacement ventilation with inflow of air from circumferential floor openings at the rear edge of the stage (5000 m^3^/h) and through slits between movable stage elements on the entire stage surface (1000 m^3^/h). This results in at least 70 m^3^/h of fresh air supply per person present on stage under the selected experimental conditions. To assess the influence of different seat distances between musicians on stage on indoor air quality, measurements were made under conditions such as those proposed as occupational health and safety measures during the pandemic, as well as under the typical conditions of a full orchestra prior to the pandemic. A total of six measurements were carried out during the repeated performance of the fantasy overture Romeo and Juliet by Pyotr Ilyich Tchaikovsky with three different seating arrangements of the orchestra. For a narrow seating arrangement (81 musicians), the distance between the strings was 1.0 m and between the wind players 1.5 m, for a medium seating arrangement (81 musicians) 1.5 m and 1.5 m, and for a wide seating arrangement (57 musicians) 1.5 m and 2.5 m, respectively (Figure 1).

### 2.1. Measurement of Indoor Air Parameters

As an indicator for fresh, unpolluted air on the stage during music making, trends of temperature, relative humidity and CO_2_ were measured at 28, 14 and one central position, respectively. Therefore, at the position of the particle counters (Figure 1), the temperatures in the floor area (2–10 cm above the floor) and above the seating position (170–180 cm above the floor) as well as the relative humidity (RH) above the seating position were also recorded with data loggers (LogTag Trix-8 data logger; LogTag HAXO-8 data logger (CiK Solutions GmbH)). The nominal accuracy of the data loggers is given as ±0.5 °C for temperatures and ±5% RH for humidity under typical room conditions, so that trends but not exact values were recorded with these loggers. In addition, both parameters were measured at the height of the seated position (120–125 cm), at the level of the particle counters (Appendix A). Furthermore, a continuous CO_2_ concentration measurement was taken with the Testo 480 in the middle of the stage.

### 2.2. Measurement of Particles

To determine whether particle accumulation occurs at the musicians’ inhalation position during music-making on stage, suggesting an increased risk of infection for musicians, particle measurement was performed. The particle counters were positioned between two seated musicians at head height (1.2–1.25 m) with the measuring direction towards the conductor’s desk in order to model the breathing position of a musician (Appendix A). The particle measurement was carried out with 7 calibrated laser particle counters PC 200 (Trotec, Heinsberg, Germany) using the “differential” analysis method. The counting efficiency is specified with 50% at 0.3 µm and 100% for particles > 0.45 µm. An air volume of 1 L was sucked in per minute and the particles of the size range 0.3–10.0 µm were determined.

To ensure that a possible particle increase or decrease could be detected with the equipment used, a preliminary measurement was performed (Appendix A). Particles were introduced into an empty room with an ATM 228 aerosol generator (Topas, GmbH, Dresden, Germany), then removed with an air purifier (KA-520-L, Kampann, Lingen, Germany) while the particle load was measured. Furthermore, an internal calibration of each particle counter was performed prior to measurement using a sterile filter according to manufacturer’s instructions.

Before the orchestra entered the stage and started the music performance, a measurement was conducted on the empty stage to obtain a reference value of usual particle load in indoor air under the present ventilation conditions. 

### 2.3. Statistical Analysis 

Visualization and statistical analysis were performed using R (version 4.0.3) and RStudio (version 2021.09.1) [13]. For visualization, the ggpubr package was activated [14]. In addition, the tidyverse, psych, and rstatix packages were used to perform statistic comparisons [15,16,17].

For this purpose, the particle counts measured by instrument number three were removed for the setup with wide seat spacing since only at this position persistently unrealistically high particle counts occurred, which we assumed to be a contamination event when changing instruments (Appendix A).

Two-sided pairwise Bonferroni corrected Welch-*t*-tests were performed to answer different hypothesis:The particle load in the indoor air on an empty stage is equal compared to music making in different seating distances.The particle load at the control position on stage is equal to the measuring positions within the orchestra.The particle load when making music with wide seating distance is equal to medium as well as narrow distances.

Significant results are defined as *p*.adj < 0.01.

### 2.4. Measurement of Airborne Bacteria

Furthermore, airborne bacteria were counted as indicator for the release of particle from the upper airways. The air samplers were mainly positioned in front of the different instrument sections in order to obtain mixed air samples that were representative for a specific type of musical instrument, especially in front of the wind instruments. 

Air sampling was performed with four impaction samplers (FH2, FH5 Markus Klotz GmbH, Bad Liebenzell, Germany; MAS-100 Eco^TM^, EMD Chemicals Inc., Gibbstown, NJ, USA; MBASS30 LKS100, Umweltanalytik Holbach GmbH, Wadern, Germany). Three devices sucked through 500 L air volume each and one device 250 L air volume within 5 min measuring time. The values determined with a volume of 250 L were extrapolated to a volume of 500 L for comparability. First, two measurements were taken on the empty stage and additionally two when musicians entered the stage to obtain values for the normal bacterial load and bacteria species in the indoor air. The air samplers were started immediately after the beginning of each music piece and were continuously used for 5 min as a mixed sample (T0). The air samplers were started three times during each piece of music, each time after a one-minute break in the measurement to change the culture medium (start at the end of minutes 6 [T6], 12 [T12] and 18 [T18]). Airborne bacteria were collected on Columbia agar with sheep blood (bioMérieux, Marcy l’Etoile, France) followed by anaerobic incubation at 37 °C for 72 h allowing efficient growth of the bacterial flora from the upper respiratory tract. Bacterial species were identified using MALDI TOF mass spectrometry and the MALDI Biotyper^®^ database (Bruker, Bremen, Germany). For further evaluation, detected species were categorized as staphylococci, other skin flora (*Cutibacterium* spp., etc.) as well as streptococci and other pharyngeal flora (Gram-negative anaerobes, etc.) according to the typical colonization sites of the human normal flora. Typical bacilli of the inanimate environment (e.g., *Bacillus* spp.) were not analyzed further due to their relatively rare detection.

## 3. Results

### 3.1. Measurement of Indoor Air Parameters

The temperature measurement showed a slight warming of the ambient air (max. 0.9 °C) in the floor area and above the seating positions during the presence of the orchestra musicians on stage (Appendix A). Within the measurement limits of the measurement system, no temperature gradient could be detected over the measurement distance of about 160 cm (Appendix A).

The measurement of the relative humidity showed a trend indicating continuous and effective removal of the humidity emitted by the people present on the stage. At the beginning of the measurement, when people entered the stage, the air humidity decreased by 2–3%. Afterwards, the relative humidity remained constant over the whole measurement period (Appendix A).

The continuous CO_2_ measurement at one position in the middle of the stage showed almost constant CO_2_ values in a corridor of approximately 420–480 ppm during the presence of the orchestra members in the phase with narrow and medium seat distances (Appendix A). For the phase with wide seat distances, values for the CO_2_ concentration are only available for the first 20 min because of technical issues. However, these values also did not differ from the values with close and medium seat distances, indicating that the CO_2_ concentrations are also not influenced by the seat distances.

### 3.2. Measurement of Particles

The results of the particle measurement are shown in Figure 2. First, the number of particles on the empty stage was determined. Here, a median particle amount of 482, 124, and 24 P/dm^3^ was detected for particle sizes 0.3–0.5 µm, 0.5–1.0 µm, and 1.0–2.5 µm, respectively. The median values of particle number at the 13 measuring positions on stage during music making were in a similar range for all three seating distances (Appendix A). However, a wide range of variation and many outliers could be observed for all measurements. 

A measuring position next to the conductor was carried along as a control with the particle counter’s suction direction approximately to the empty audience. At this position, very stable concentrations could be measured and movement of musicians during music making has barely any influence. 

The statistical analysis of the particle measurement showed that the particle concentration on the empty stage was similar to the particle amount during music making at wide, medium, and narrow seating distances. 

For all three seating arrangements studied, a statistical test was performed to determine if the particle count at the control position next to the conductor differed from the mean of all measurement positions within the orchestra. In the majority of cases, a statistically significant difference was found. 

A comparison was also made between the different seating arrangements. The mean of the measured particle counts was statistically significantly different, except for particle sizes 0.5–1.0 µm and 1.0–2.5 µm when comparing close versus medium seat spacing (Figure 2, Appendix A).

### 3.3. Measurement of Airborne Bacteria

Position-dependent differences were observed for the determination of the number of airborne bacteria and the species of the detected microorganisms. A total of 2287 individual bacterial colonies were examined by mass spectrometry. Moreover, 862 isolates belonged to the genus *Staphylococcus* and 744 isolates could be assigned to the other typical skin flora species (mainly *Cutibacterium* spp.). A further 528 isolates were identified as typical representatives of the pharyngeal flora. Only 21 isolates were categorized as environmental bacteria. For 132 isolates, no clear identification could be achieved. 

In order to evaluate the influence of music performance on stage on the bacterial load of the room air and the possible changes in prevailing bacterial species, measurement of the basic bacterial load and bacteria species in the air was carried out on the empty stage (data not shown). At all positions, only very low numbers (<20 colony-forming units [CFU]/500 L air volume) of bacteria and only species that can be assigned to the human skin flora could be detected. In addition, a comparative measurement of the bacterial load of the air was performed during the phase when the musicians entered the stage (data not shown). This phase showed an increase in the total number of bacteria with a predominant proportion of species belonging to the skin flora.

While playing the piece of music, distinct differences in the relative exposure of the air to skin and pharyngeal flora at different measurement positions in front of different instrument groups were observed (Figure 3). Relatively low bacterial counts were measured in the air in front of the string instruments compared to the mere presence of people on stage. These bacteria were predominantly assigned to the skin flora. At the position in front of the flutes (Figure 3C), a detection rate comparable to the mere presence of people on stage was observed for all three seating arrangements. At the position in front of the oboes (Figure 3D), a higher detection rate was observed for the different measurement setups on stage. The increased detection rate at this position was mainly due to the detection of skin flora. More bacteria were also detected in the air in front of the other wind instrument groups studied (clarinet, bassoon, trombone, trumpet) than in the mere presence of people on stage. For these instrument groups, a relative increase in the proportion of pharyngeal flora species was identified in most measurement phases, which was most pronounced in the position in front of the clarinets (Figure 3E). No clear correlation with seat spacing was observed for either the increase in the total number or the relative increase in the proportion of pharyngeal flora. It can be assumed that the seat distances have no influence on the respective position-dependent and probably instrument-group-typical detection rates. 

Furthermore, differences were determined depending on the score of the piece of music being played. Almost all instrument groups showed a lower detection rate of bacteria at the beginning of the piece of music (time T0, minutes 1–5) as in the other phases. In the course of the second measurement phase (time T6, minutes 7–11), up to the third measurement phase (time T12, minutes 13–17), an increasing tendency was shown. Especially in the third measurement phase (T12) a clear increase of pharyngeal flora can be observed for the clarinets, bassoons, trumpets and trombones. This observation is compatible with the score of the piece of music, which has a musical highpoint here and is represented by loud dynamics, many tutti passages, as well as a high participation of the wind instruments.

## 4. Discussion

The collected measurement data show a very good indoor air quality during all measurement phases, even with narrow seating distances while the orchestra is present on stage. The almost constant values of the CO_2_ concentration are compatible with 100% fresh air proportion in the supply air and a high air exchange rate at the same time. The German Technical rules for workplaces issued by the German Federal Institute for Occupational Safety and Health specify a limit of 1000 ppm for CO_2_ concentration indoors and recommend that a value of 800 ppm be maintained during the SARS-CoV-2 pandemic [6]. During the entire measurement period, the measured value was well below 800 ppm, indicating a high air exchange. The reduction of CO_2_ concentration in the air is associated with an infection prevention effect [18].

The median values of the measured particle load are consistently at a very low level, for particle sizes 0.5–1.0 µm comparable to rooms with an ISO 7 classification [19]. Furthermore, they are in a similar range compared to control measures without musicians on the stage. On average, no increase in particle load above the background noise of the present room could be detected. Differences were observed both when comparing the measured particle counts of different seat distances and when comparing them with the respective control position. Based on measurement uncertainties, large standard deviations, and a smaller number of data for the control position, it can be assumed that these differences have no measurable influence on the infection risk for the musicians, since the particle counts as a whole do not increase above the values of the blank measurements.

Particle loads described in the literature are not directly comparable with the particle concentrations measured in this study due to the different measurement methods. In addition, the basic particle load of rooms varies between different studies. 

The locally observed short-term increase in particle concentration, especially for the size group 2.5–5.0 µm (Appendix A), could also be caused to a relevant extent by the resuspension of fine dust particles (e.g., rosin) and not by the release of aerosols. The increases in particle concentration correlate with musical highpoints and emotional playing, associated with increased movements of the musicians. In addition, it can be clearly deduced that slight increases in particle load in the air, which occurred after increased movement, returned to a low level within 3–5 min. The obtained data indicate that released particles or aerosols are removed very quickly by ventilation and that there is no accumulation on the stage at any time. 

In the area of the examined stage, active ventilation is carried out by the principle of displacement ventilation. With this ventilation technique, supply air is introduced close to the floor with a low flow velocity and a slightly lower temperature compared to the ambient temperature. Due to the heating of the air by the people present the air rises vertically, which enables the efficient removal of polluted air, especially at very high room heights. In low rooms, an excessively high temperature gradient can lead to the formation of a barrier layer and the accumulation of polluted air in the breathing zone of people [20,21]. A temperature difference of 0.4–0.5 °C per meter of height is assumed to be a critical temperature gradient, which could cause the stratification height to fall into the inhalation zone of persons [22]. During the measurements carried out, no significant temperature difference was observed in the measurement of temperature near the floor and above seated orchestra members, such that the rapid removal of exhaled air and aerosols is very efficient. In the investigated setting, fresh air quantities are introduced, which result in a supply of more than 70 m^3^/h for all seating positions. It can therefore be assumed that in the breathing zone of persons present, unpolluted fresh air continuously flows in from the source air outlets. In a study on a mathematical risk assessment model with a comparison of different prevention measures, it was shown that the person-related air flow rate per hour of stay is a favorable indicator for evaluating the preventive effect of ventilation measures in infection prevention [18]. In addition, a previous experimental study for displacement ventilation showed that people who stand with their backs to the person exhaling are not exposed to the contaminated air of the latter [23]. 

Cultural detection of human bacterial flora with assignment of identified species to typical body regions might be a suitable measurement method to model exposure to potentially infectious particles from the upper respiratory tract. However, no comparable data for the quantitative release of droplets or pharyngeal flora can be found in the literature yet. 

The detection rate of culturable bacteria in the air provides no evidence of an accumulation of potentially infectious droplets or an adverse effect of reduced seating distances. Even for the peak values of pharyngeal flora, the relative number of detected bacteria was low if the average respiratory minute volume of humans (approximately 8 L/min) is taken into account compared to the very high intake volume for airborne bacteria measurement (100 L/min). It has been demonstrated that musical instruments may be contaminated with typical pharyngeal and environmental flora [24]. Thus, the release of pharyngeal flora could be due in part to residual contamination of the instruments. Within the framework of the currently ongoing SOBADRA study (preprint consensus statement [25], it was shown that pharyngeal flora is contained in droplets produced by respiratory activities and that microorganisms can thus be transmitted via this route. However, the authors consider the transmission route via droplets released during music-making to be negligible.

Due to the need for personal operation of the air sampler during music making, the measurement of the bacterial load in the air must be carried out in smaller distances as the seating distance, which results in a slight overestimation of the bacterial load in the air compared to the load in the breathing zone of musicians. The assignment of the bacterial species measured in front of the instrument groups to their probable site of origin, can support a further classification of instruments into the categories low, normal and high risk. For the clarinets classified as normal or high risk in different literature [4,6], the relatively high proportion of pharyngeal flora is an indicator to classify this instrument into the high-risk group. Interestingly, for oboes, which were also classified as normal or high risk in different studies, a significantly different situation emerged. The detected number of bacteria per air volume was very comparable to the number of bacteria observed in front of the group of clarinets. However, the low relative proportion of pharyngeal flora should be taken as an indicator that these instruments should more appropriately be classified in the normal or low risk group. The classification of trombones and trumpets into the group of high-risk instruments is adequate due to the high relative proportion of pharyngeal flora detectable in individual tips. The classification of flutes in the group of normal risk instruments is appropriate to the overall lower total load of bacteria in air in front of this instrument group, in comparison to other wind instruments.

A very high detection rate of skin bacteria in front of the second violins during the first measurement setup is most likely an environmental contamination (e.g., dust accumulation) in the vicinity of this measurement position, which dissipated in the course of the measurement day (e.g., due to remodeling measures). 

When classifying the groups, it should be noted that the terminology of high-risk instruments chosen by other working groups must be seen in the overall context. The relative amount of viral particles in the upper airways can vary over a wide range [26], and not every person suffering from a viral infection will produce infectious aerosols. In the case of the observed effective ventilation without accumulation of particles on the stage, no particularly increased risk of virus spread can be assumed for the infection of a musician with a low or moderate viral load, even when playing an instrument of the so-called high-risk group. In particular, there were no differences in the relative exposure depending on the seat spacing or the number of instruments in the respective instrument group on stage. Under the conditions observed, the exclusion of so-called super spreaders with extremely high viral loads in the upper respiratory tract by means of suitable test concepts is therefore a sufficient way to achieve a high level of occupational safety for musicians.

The following limitations of the present study should be noticed. The release of particles and pharyngeal flora during music making also depends on the playing time (playing times compared to break times) of the respective instruments. The playing proportions of instrument groups in musical pieces vary, such that a locally increased rate of pharyngeal flora and particles can also be caused by a large playing time proportion of certain instruments within the piece. Further studies on the release of pharyngeal flora per instrument group with standardized playing time may therefore be informative in order to evaluate this influence more precisely. It should also be taken into account that during each of the experiments, measurements were taken in front of a group of instruments, so the distance of each musician to the measuring instruments varied to some extent. For some instrument groups, this measurement position may not correspond directly to the main flow direction of the emitted air when playing music. Therefore, the number of particles and germs detected in the air may possibly be underestimated for individual instruments. The aim of the study was to evaluate whether there could be an accumulation effect of particles and pharyngeal flora during music making as an indicator of increased risk of infection for the entirety of the musicians on stage. Viral respiratory infectious agents are usually emitted adherent to larger mucosal particles and not alone [27]. Furthermore, Kriegel et al. identified particles between 0.3 and 5.0 µm as the most important carriers of viral pathogens [18]. Therefore, the particle counters with a measurement limit of 0.3–0.5 µm used here, as well as the impaction collectors, thus had a suitable accuracy and measurement range to be able to evaluate this issue. Due to the positioning of the devices, no statement can be made about the particle release of individual musical instruments, but it is possible to make a statement about the distribution in the entire orchestra.

Finally, it should also be noted that the combined method for evaluating the release of particles and pharyngeal flora was used for the first time in this study. Additional studies should be conducted in the future to better understand the significance of detecting pharyngeal flora during music making. Before a general application of this method, it should be tested in different locations with different ventilation principles.

## 5. Conclusions

A highly effective ventilation system based on displacement ventilation results in no aerosol accumulation on the stage, regardless of the seating arrangement or the number of musicians playing music, when an orchestra is present. In contrast to the quantitative measurement of particles or aerosols, the detection of bacterial flora from the upper airways might be an important additional aspect for the individual assessment of potential infection risk in large groups of persons. With suitable test concepts to exclude highly positive persons as well as infection prevention concepts to avoid droplet infections, a very high safety level for orchestra members can be reached by optimized ventilation, even with narrow seating.

## Figures and Tables

**Figure 1 ijerph-19-09939-f001:**
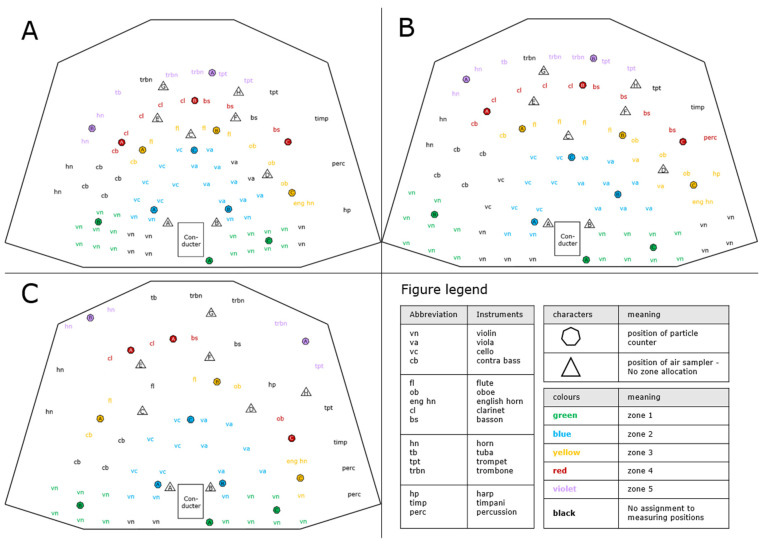
**Measurement setup on stage.** The different seating positions of the musicians in the narrow (**A**), medium (**B**) and wide (**C**) arrangements are shown. The abbreviations represent the different instruments (see legend). The positions of particle counters and air samplers are displayed by a hexagon and a triangle, respectively. Furthermore, for comparison and interpretation of results, the measuring positions of particle counting were assigned to zones, which is indicated by the different colorings of the devices.

**Figure 2 ijerph-19-09939-f002:**
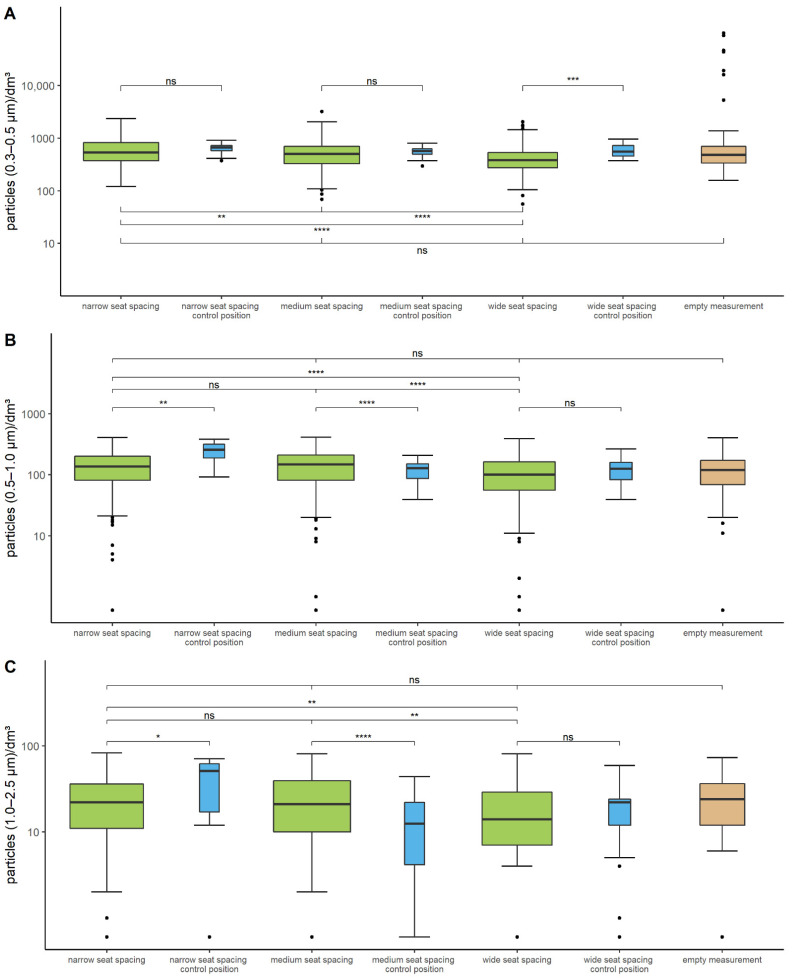
**Particle count in indoor air during music making.** Shown is the number of particles in indoor air for particle sizes 0.3–0.5 µm (**A**), 0.5–1.0 µm (**B**), and 1.0–2.5 µm (**C**). The detection limit for all particle sizes is 1 particle/dm^3^. The width of the boxplots represents the number of contained data. The green colored boxplots display the number of particles during music making in the different setups of seating (narrow, medium, wide) on stage. Blue colored boxplots represent the number of particles at the control position on stage as well as orange-colored boxplots the empty measurement. Brackets display the results of the statistical analysis. The characters “ns”, “*”, “**”, “***” and “****” represent the adjusted *p*-value “not significant”, “<0.01”, “<0.001”, “<1 × 10^−4^”, “<1 × 10^−5^”, respectively.

**Figure 3 ijerph-19-09939-f003:**
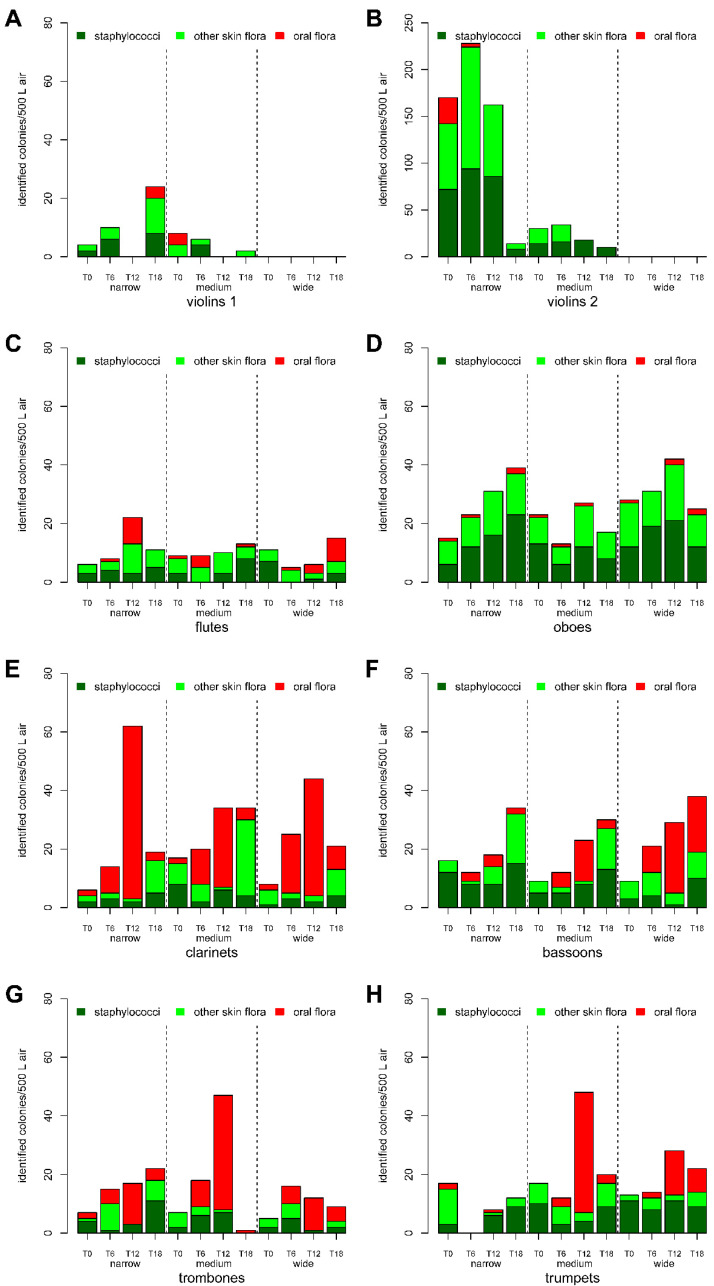
**Detection of bacteria in front of different instrument groups** as a function of time. The total number of staphylococci (dark green), other skin flora (light green) and pharyngeal flora (red) in front of the 1st violins (**A**), 2nd violins (**B**), flutes (**C**), oboes (**D**), clarinets (**E**), bassoons (**F**), trombones (**G**) and trumpets (**H**) at narrow, medium and wide seat distances at time points T0, T6, T12 and T18 are displayed, respectively.

## Data Availability

All data are available in the main text or the Appendix A.

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
