# Peer review of "Reading the Score of the Air—Change in Airborne Microbial Load in Contrast to Particulate Matter during Music Making"

_ijerph, 2022, doi:10.3390/ijerph19169939_

Round 1

Reviewer 1 Report

The manuscript “Reading the score of the air - How does the air quality change on the stage when an orchestra plays?” describes measurements for several air quality parameters in a full orchestra. The measurements aim at the exposure risk of the musicians against air pollutants (e.g. airborne particles). The study has been performed in the light of the COVID pandemic but is of scientific interest beyond this topic. Some aspects need to be clarified before publication.

Indoor air quality measurements require a complete description of the indoor environment. Else, the evaluation of the measured concentrations is not possible. The authors need to describe the size of the room and the ventilation condition. The text indicates that the experiment has been performed in a full-scale orchestra hall (e.g. Sendesaal Hannover) with a ventilation rate of 70 m³/h/p. Due to the fact that the room has displacement ventilation and is not a mixed system the measured values are not easy to interprete. If the authors have information about the true air exchange rate (e.g. decay of a tracer gas) or an estimated air exchange rate this would be a valuable addition to the manuscript.

The experimental setup has been described well but the selection of the measurement positions is not clear from the text. Figure 1 indicates an even distribution of the instruments in the different setups. In setup C some instruments were right at the edge of the stage. It is not clear where the ventilation ducts on the stage are placed. Are they also evenly distributed over the stage or at the edge of the stage. In the latter case, there should be a gradient from the edge to the center of the stage. The results of the CO2 measurements could be useful indicators here. The results are, however, not shown in the manuscript or the SI. It is mentioned that the CO2 values ranged from 420 to 490 ppm on stage. These values are extremely low (the global atmospheric CO2 level in 2021 was approx. 415 ppm (https://gml.noaa.gov)). Considering the fact that approx. 80 persons were in the room this would indicate extreme ventilation conditions. This extreme ventilation condition is underlined by the presented particle plots (Figure 2). These plots show very large variation in the measured concentrations which nearly looks like instrument noise. The authors need to perform a statistical analysis of these values. Some peaks look like true particle events but this is hard to see on a log-scale. If the values represent the full population, I recommend using a box-plot description instead. Perhaps, a differentiation between the different position can be observed then.

The authors should also mention any blank experiments they made prior or after the musicians on stage.

The authors should discuss at one point the expected nature of the particles. Ref. 4 indicates that the particles from wind instruments are in the range of 2-3 µm. It can be presumed that these water droplets undergo rapid changes at such heavy ventilation conditions. Figure 2 A indicates that the highest number of particles are in the range of 300 nm and below. A brief description of the expected sources for these fine particles would be helpful to estimate the relevance of these particles and possible source-oriented mitigation strategies. If the ventilation is very strong, the presence of 10^4 #/L particles could indicate a very strong source.

The authors should also revise the conclusions of the study. In principle, the authors suggest the use of “personal ventilation” techniques for each musician (see e.g. Cermak, R.;  Melikov, A. K.;  Forejt, L.; Kovar, O., Performance of personalized ventilation in conjunction with mixing and displacement ventilation. Hvac&R Research 2006, 12 (2), 295-311, and many more). Since the description of the ventilation on the used stage is incomplete (see above) it is not clear if this true for many different setup or just this one situation. Thus, the authors should provide an estimation of the transferability of their results here. Further, the conclusion that particle measurements are less effective than biological indicators for assessing of infection risks can not entirely be answered from the measurements. Since direct optical techniques were used (which exclude the ultra-fine particle fraction) it is not clear if smaller particles might be a better indication instead. Furthermore, the authors compare online measurement techniques and accumulation techniques here. A cumulative analysis (e.g. regular filter-based PM2.5/PM10 measurements) might provide much better results than the optical sensor techniques.

Minor aspects:

 During typesetting some errors (e.g. hyphon in line 29) occurred. The “author contributions” as well as the text for Appendix A and B are template text blocks that should be updated/removed.

Reviewer 2 Report

Although it may seem a little useful topic for the purposes of a publication in a newspaper that concerns the environment and health together, the topic and the need to investigate the highest number of aggregation systems together, by virtue of the recent Covid-19 epidemic , make a paper on such a topic acceptable.

The research as a whole is well presented, the introduction exhaustive, the materials and methods used, as well as the results, are well structured. Sometimes carbon dioxide reports the 2 incorrectly not in subscript.

The article is not particularly bright but acceptable in this form.

Author Response

Point 1: Sometimes carbon dioxide reports the 2 incorrectly not in subscript.

Response 1:.The errors mentioned have been removed.

Reviewer 3 Report

Comments to the Author
Major comments:

(1) For the abstract (line 10-24), can the authors provide an additional statement in the end about what action should be taken with the present findings. In line 10-19, can the authors refine the sentences further? The idea of abstract is about to present your key findings in the study but not about repeating contents in the introduction.

(2) In line 60, can the authors provide addition statements about the aims of your study?

(3) In line 61-126, can the authors divide this part to different sub-sections as the entire section is too long at present. Can the authors further provide an additional section about quality control/assurance?

(4) In Section 3.0 (line 127-220), under line 139 and 201-202, can the authors please state further how the conclusions about associations and correlations could be reached in the analysis. Did any statistical analysis perform in these circumstances?

(5) In discussion (line 221-344), under line 231-238, can the authors provide more citations to support the claims? In line 285-286 and 287-304, please provide additional information to support your statement.

Minor comments:

1.     Title, line 2-3

Please revise your title. A standard one without “?” will be more appreciated.

2.     Keywords, line 25
Please use indoor air quality and particulate matter instead.

3.     Figure 2, line 144

These trends in the three different colors are hard to interpret.

4.     Section 3, line 164

Please use approximately instead.

5.     Section 1-5, line 27-353

Please revise the caption style  (from suppl. fig. 1 and fig. 1 to Supplementary Materials: Figure 1 and Figure 1) throughout the entire manuscript.
